# Pilot Study of the Effects of Chronic Intracerebroventricular Infusion of Human Anti-IgLON5 Disease Antibodies in Mice

**DOI:** 10.3390/cells11061024

**Published:** 2022-03-17

**Authors:** Sara Alvente, Gabriele Matteoli, Laura Molina-Porcel, Jon Landa, Mercedes Alba, Stefano Bastianini, Chiara Berteotti, Francesc Graus, Viviana Lo Martire, Lidia Sabater, Giovanna Zoccoli, Alessandro Silvani

**Affiliations:** 1Department of Biomedical and Neuromotor Sciences, University of Bologna, 40123 Bologna, Italy; sara.alvente2@unibo.it (S.A.); gabriele.matteoli2@unibo.it (G.M.); stefano.bastianini3@unibo.it (S.B.); chiara.berteotti@unibo.it (C.B.); viviana.lomartire2@unibo.it (V.L.M.); giovanna.zoccoli@unibo.it (G.Z.); 2Hospital Clínic, Institut d’Investigacions Biomediques August Pi i Sunyer (IDIBAPS), 08036 Barcelona, Spain; lmolinap@clinic.cat (L.M.-P.); landa@clinic.cat (J.L.); mealba@clinic.cat (M.A.); francesc.graus@idibaps.org (F.G.); lisabate@clinic.cat (L.S.); 3Alzheimer’s Disease and Other Cognitive Disorders Unit, Neurology Service, Hospital Clínic, IDIBAPS, 08036 Barcelona, Spain; 4Neurological Tissue Bank, Biobanc, Hospital Clínic, IDIBAPS, 08036 Barcelona, Spain; 5Centro de Investigación Biomédica en Red Enfermedades Raras (CIBERER), 46010 Valencia, Spain

**Keywords:** IgLON5, mice, animal models, Tau, sleep, breathing, motor control, licking

## Abstract

Background: Anti-IgLON5 disease is a rare late-onset neurological disease associated with autoantibodies against IgLON5, neuronal accumulation of phosphorylated Tau protein (p-Tau), and sleep, respiratory, and motor alterations. Purpose: We performed a pilot study of whether the neuropathological and clinical features of anti-IgLON5 disease may be recapitulated in mice with chronic intracerebroventricular infusion of human anti-IgLON5 disease IgG (Pt-IgG). Methods: Humanized transgenic hTau mice expressing human Tau protein and wild-type (WT) control mice were infused intracerebroventricularly with Pt-IgG or with antibodies from a control subject for 14 days. The sleep, respiratory, and motor phenotype was evaluated at the end of the antibody infusion and at least 30 days thereafter, followed by immunohistochemical assessment of p-Tau deposition. Results: In female hTau and WT mice infused with Pt-IgG, we found reproducible trends of diffuse neuronal cytoplasmic p-Tau deposits in the brainstem and hippocampus, increased ventilatory period during sleep, and decreased inter-lick interval during wakefulness. These findings were not replicated on male hTau mice. Conclusion: The results of our pilot study suggest, but do not prove, that chronic ICV infusion of mice with Pt-IgG may elicit neuropathological, respiratory, and motor alterations. These results should be considered as preliminary until replicated in larger studies taking account of potential sex differences in mice.

## 1. Introduction

Anti-IgLON5 disease is a rare (ORPHA:420789, www.orpha.net, last accessed on 14 March 2022) late-onset (median 62 years [1]) neurological disease associated with autoantibodies against IgLON5 [2], a cell-surface protein of unknown function [3] preferentially expressed in neurons [4]. The clinical spectrum of anti-IgLON5 disease includes prominent sleep and motor alterations [2]. Patients show episodes of undifferentiated or poorly structured non-rapid-eye-movement (NREM) sleep with excessive muscle activity, rapid-eye-movement (REM) sleep behavior disorder, obstructive sleep apneas, and stridor [5]. Nocturnal activity is strikingly increased in the face of a moderate reduction of total sleep time and sleep efficiency [5]. Motor dysfunction includes dysphagia, abnormal orofacial movements, ataxia, gait instability, and dysautonomia [1,6,7,8]. Initial post-mortem studies showed a characteristic pattern of neuronal loss and gliosis, with neuronal accumulation of phosphorylated three-repeat (3R) and four-repeat (4R) Tau protein isoforms mainly in the hypothalamus and the brainstem tegmentum [9]. In two subsequent reports, Tau deposits were mostly detected in the hippocampus, suggesting a primary age-related hippocampal tauopathy, with relative brainstem sparing [10,11].

Anti-IgLON5 disease has sparked interest as a possible link between autoimmunity and neurodegeneration [12]. However, the causal role of anti-IgLON5 antibodies is still unclear. Building upon the study of paraneoplastic and myasthenic disorders, recent research has expanded the concept that neurological diseases can be mediated by neuronal autoantibodies [13], with inflammation also playing a role in pathogenesis (cf., e.g., [14,15,16,17]). Anti-IgLON5 disease is linked with HLA-DRB1*10:01 and HLA-DQB1*05:01 alleles [1], and may respond to immunotherapy [18]. However, HLA allele positivity is not associated with a better response to immunotherapy; rather, limited evidence suggests better response in patients carrying HLA-DQB1*05:01 without HLA-DRB1*10:01 [18]. Nevertheless, revisitation of Witebsky’s postulates to define criteria for autoimmune diseases indicate that association with HLA haplotype and response to immunotherapy both represent circumstantial evidence suggesting autoimmunity [19]. According to these criteria, demonstration of antibody pathogenicity should include in vitro damage to cells carrying the IgLON5 antigen and in-vivo duplication of anti-IgLON5 disease in experimental animals by administration of human antibodies [19]. In vitro evidence is accruing in support of this hypothesis: anti-IgLON5 antibodies from patients with anti-IgLON5 disease (Pt-IgG) cause irreversible IgLON5 protein internalization [20] and cytoskeletal disruption [21] in cultured rat hippocampal neurons, and phosphorylated Tau protein (p-Tau) deposition and cell death in neurons derived from human induced pluripotent stem cells [22]. On the other hand, in vivo evidence is lacking, likely due to the complexity of evaluating the results of administration of human antibodies to experimental animals and to the number of potential experimental parameters to explore without prior knowledge. These parameters include the species, sex, genetic background, and age of the experimental model, the dose, administration route, and duration of antibody infusion, the specific phenotypic and neuropathological outcome measures, and the time lag between antibody infusion and the expected phenotypic or neuropathological changes.

We aimed to perform a pilot study of whether chronic intracerebroventricular (ICV) infusion of Pt-IgG in mice recapitulates the clinical and neuropathological features of anti-IgLON5 disease. We performed experiments both in hTau mice, which show a spontaneous tendency to p-Tau deposition and late-life neurodegeneration due to the expression of 3R and 4R human Tau protein instead of murine Tau [23,24], and in C57Bl/6J wild-type (WT) mice, which are arguably the most widely studied inbred mouse strain.

## 2. Materials and Methods

### 2.1. Animals

The experimental study protocol complied with the EU Directive 2010/63/EU for animal experiments and was approved by the Committees on the Ethics of Animal Experiments of the University of Bologna and of the Italian Ministry of Education, University, and Research (protocol code 780/2017-PR, approved 16 October 2017). Experiments were performed on C57Bl/6J wild-type (WT) mice and on congenic hTau mice [23] (B6.Cg-Mapt^tm1(EGFP)Klt^ Tg(MAPT)8cPdav/J, the Jackson Laboratory stock n. 005491, www.jax.org, last accessed on 17 March 2022) with pure C57Bl/6J background (>10 generations of backcrossing). The hTau mice are homozygous for a null mutation of the mouse *Mapt* gene, which codes for the murine Tau protein, and hemizygous carriers of a transgene that includes the human *MAPT* gene, which codes for all six isoforms of the human Tau protein [23].

All mice were bred at the Department of Biomedical and Neuromotor Sciences of the University of Bologna, Italy. The hTau mice were obtained by mating male hTau founder mice with female founder mice that were homozygous for the null mutation of the mouse *Mapt* gene and non-carriers of the human *MAPT* transgene. Founder mice were purchased from the Jackson Laboratory through Charles River Italy, Calco, Italy. The genotype of the progeny was assessed by polymerase chain reaction on ear tissue biopsies, using protocol and primers as suggested by the Jackson Laboratory.

We focused on female hTau (*n* = 12) and WT (*n* = 8) mice because evidence at the time of study design indicated that brainstem p-Tau deposition was greater in female than in male hTau mice already at 4 months of age [25]. As contrasting results on this issue were subsequently published [26,27], we included a replication group of 7 male hTau mice, also considering the lack of gender-specificity of human anti-IgLON5 disease [2].

All mice were group-housed under a 12:12-h light–dark cycle with ambient temperature set at 23 °C and free access to water and food (4RF21diet; Mucedola, Settimo Milanese, Italy). Mice employed in the experiments were housed singly from the time of surgery to prevent accidental damage by conspecifics.

### 2.2. Antibody Purification from Patients with Anti-IgLON5 Disease and Control Subjects

The Ethic Committee of Hospital Clinic, Barcelona, approved the study (R091217–12). Patients or proxies gave consent for the storage and use of serum, cerebrospinal fluid, and clinical information for research purposes. Informed consent was obtained from all subjects that provided antibodies employed in this study. Total G immunoglobulins (IgG) from sera of a patient with anti-IgLON5 disease (Pt-IgG) and of a control subject without the disease (Ctrl-IgG) were obtained and purified at the Institut d’Investigacions Biomèdiques August Pi i Sunyer (IDIBAPS), Hospital Clínic, Universitat de Barcelona, Barcelona, Spain. The patient was a female aged 65 years with a typical clinical pattern, predominant bulbar phenotype, and chronic evolution, with positivity to HLA susceptibility alleles and IgG4 predominance. The control subject was matched for age and gender. Briefly, 1 mL of serum was incubated with 0.5 mL of protein A/G sepharose beads. Purification was performed with columns of protein A/G sepharose beads (Thermo Fisher Scientific, Waltham, MA, USA) for 30 min, eluted with 100 mM glycine, pH 2.5 and dialyzed against phosphate buffered saline (PBS). The concentration of total IgG was 2 mg/dL, which was well tolerated in mice by ICV infusion in a previous study on human anti-N-methyl D-aspartate (NMDA) receptor antibodies [28].

The specificity of the binding of purified IgG was assessed on transfected HEK cells expressing human IgLON5 and on rat brain tissue as previously reported in detail [2], based on the 100% and >98% identity of IgLON5 sequences between rats and mice and between mice and human subjects, respectively (NCBI Blast sequences XP_218634.5, NP_001157990.1, and NP_001094842.1 in Rattus norvegicus, Mus musculus, and Homo sapiens, respectively). In particular, the antibodies were immunoabsorbed with live HEK293 cells expressing IgLON5 or with mock transfected cells. After six rounds of immunoabsorption, the antibodies were applied to sections of rat brain and the reactivity developed as described.

### 2.3. Overview of the Experimental Protocol

We performed surgery and started ICV antibody infusion in young adult mice at 19–22 weeks of age, when p-Tau deposits in hTau mice are already detectable but not yet accompanied by cell loss [23,24,29]. ICV infusion was performed continuously for 14 days with implanted osmotic minipumps. Sleep, respiratory, and motor phenotype were evaluated twice for each mouse, the first time at the end of the 14-day ICV antibody infusion (d14), and the second time at least 1 month after the end of the infusion (d45) to account for possible delayed effects. Mice were eventually sacrificed by trans-cardiac perfusion with fixative (4% paraformaldehyde) under anesthesia overdose (isoflurane 4% in O_2_). This procedure occurred after d45 recordings except for 5 mice (1 female WT with Ctrl-IgG 1 male and 3 female hTau mice with Pt-IgG), in which experiments were interrupted at 36 ± 1 days after surgery due to partial detachment of the head implant. Overall, mice were aged 27.7 ± 0.5 weeks at death. An overview of the experimental protocol is shown in Figure 1.

### 2.4. Surgery and ICV Antibody Infusion

Twenty-four hours before surgery, two osmotic pumps per mouse (model 1002, Alzet, Cupertino, CA, USA; volume: 100 μL; flow rate: 0.25 μL/h; duration of infusion: 14 days) were each loaded with 100 μL of either Pt-IgG or Ctrl-IgG for different mice. The pumps were then connected to a 0.28 mm (internal diameter) vinyl tube (C314CT, PlasticsOne, Roanoke, VA, USA) and left overnight in sterile saline at 37 °C to ensure that nominal pump flow could be reached immediately upon implantation. The total nominal pump flow of 0.5 μL/h was negligible with respect to the estimated cerebrospinal fluid production rate in mice (22.2 μL/h), thus minimizing the expected increases in intracranial pressure [30]. The 14-day duration of the antibody infusion was shown effective in a previous study on human NMDA receptor antibodies [28].

Mouse weight at surgery was 23.3 ± 0.4 g, 21.4 ± 0.3 g, and 28.4 ± 0.3 g in female WT and in female and male hTau mice, respectively (means ± SEM). Mice underwent surgery under isoflurane anesthesia (1.8–2.4% in O_2_) with intra-operative analgesia (Carprofen 0.1 mg subcutaneously) and antibiotic prophylaxis (benzylpenicillin benzathine, 12,500 IU/kg, and dihydrostreptomycin sulphate, 5 mg/kg, subcutaneously). Mice were placed in a stereotaxic frame, and a bilateral cannula (PlasticsOne, model 3280PD-2.0/SP) was inserted into the cerebral ventricles (0.6 mm posterior and 1 mm lateral to bregma, at a depth of 2 mm). Each arm of the cannula was connected by a catheter to one osmotic pump, which was implanted subcutaneously on the back together with the catheter [28]. Mice were also implanted with two miniature stainless steel screw electrodes (2.4 mm length, PlasticsOne) in contact with the dura mater through burr holes in the frontal and parietal bones (2 mm anterior and 2 mm to the right of bregma and lambda, respectively), to record a differential electroencephalogram (EEG) signal, and with two Teflon-coated stainless steel wires (Cooner Wire, Chatsworth, CA, USA) with a 5 mm bare segment inserted in the nuchal muscles, to record a differential electromyogram (EMG) signal. The electrodes were connected to a miniature electrical socket, which was secured to the skull with dental cement together with the cannulas as previously described [31,32].

At the end of the 14-day infusion period, the pumps were surgically removed (Figure 1) with a brief surgical procedure under general anesthesia to prevent damage to the mice due to leakage of pump hyperosmotic fluid as suggested by the vendor. The catheters that connected the pumps to the cannulas were left in place in the dorsal subcutaneous tissue after careful sealing of their distal ends.

### 2.5. Neuropathological Examination

Brains were removed, postfixed by immersion in 4% paraformaldehyde for 1 h at 4 °C, cryoprotected with 40% sucrose for 48 h, and embedded in paraffin. The analysis of p-Tau was performed by immunohistochemistry (IHC) on 4-µm paraffin sections, which were placed in a BOND-MAX Automated IHC Stainer (Leica Biosystems Melbourne Pty Ltd., Melbourne, Australia). Tissues were deparaffinized and pre-treated with the Epitope Retrieval Solution 1 (citrate buffer, pH 6) at 98 °C for 50 min. The monoclonal antibody against p-Tau (Ser202, Thr205: AT8, Thermo Fisher Scientific, Waltham, MA, USA) was incubated at 1/1000 concentration for 20 min. Subsequently, tissues were incubated with polymer for 8 min and developed with DAB-Chromogen for 10 min. Slides were counterstained with hematoxylin.

Based on the pattern of AT8 staining particularly in the hippocampus and brainstem, mice were classified in two groups corresponding to higher vs. lower AT8 staining. These procedures were performed in the laboratory of neuroimmunology at the Institut d’Investigacions Biomèdiques August Pi I Sunier, Barcelona, Spain by three expert investigators blind to the source (Pt-IgG vs. CTRL-IgG) of the antibodies received by each mouse.

### 2.6. Assessment of Daily EMG Activity Profiles

Both at d14 and at d45, mice underwent 48-h continuous EEG and EMG recordings while freely behaving in their own cages, which were equipped with an electrical swivel and a balanced suspensor arm [31,32]. The EMG root mean square (rms) values over 4-s epochs were normalized to the minimum and maximum values of their distribution over the 48-h recordings after trimming the bottom 0.5% and top 0.5% of the EMG values, which are potentially artefactual [33]. The resulting normalized EMG values ranged from 0 to 100 and were averaged over 2-h bins for plotting and over 12-h light and dark periods for statistical analysis.

### 2.7. Assessment of Sleep Phenotype

Both at d14 and d45, mice underwent 8-h continuous recordings of EEG, EMG, and breathing, starting at the onset of the light period, while freely behaving in a modified whole-body plethysmograph equipped with an electrical swivel, as previously described in detail [34,35]. Scoring of wake-sleep states (wakefulness, NREM sleep, and REM sleep) was performed with a 4-s time resolution and a validated procedure [36]. The sleep scoring was refined with manual review of raw EEG and EMG recordings by experienced investigators blind to the source (Pt-IgG vs. Ctrl-IgG) of the antibodies received by each mouse. Sleep architecture was assessed by analyzing the percentage of recording time spent in each wake-sleep state and the distribution of NREM sleep episode duration. The latter was based on three arbitrarily defined duration bins: 4–8 s, 12–120 s, >120 s. To assess sleep quality, the EMG rms was computed for each 4-s epoch and averaged for all 4-s epochs in wakefulness, NREM sleep, and REM sleep. The resulting EMG rms values during sleep were divided by the corresponding average values during wakefulness and analyzed. Moreover, the ratio of EEG spectral power in the theta (6–9 Hz) and delta (0.5–4 Hz) frequency ranges was computed for each 4-s epoch, and its median value was analyzed over all 4-s epochs of NREM sleep and REM sleep.

### 2.8. Assessment of Sleep-Related Breathing Phenotype

Breathing during sleep was analyzed with procedures previously described in detail [34] focusing on values of ventilatory period during sleep and on the number of apneas per hour of NREM sleep and REM sleep. We did not analyze augmented breaths (“sighs”), absent a direct translational relevance for patients with IgLON5 disease. Although sighs are often followed by apneas in mice, apneas do occur also irrespective of sighs [37].

### 2.9. Assessment of Ataxia and Gait Instability

The mouse motor function was assessed at d14 and d45 with a composite phenotype scoring system originally developed for evaluating mouse models of cerebellar ataxia [38]. Briefly, both at d14 and at d45, mice were tested three times for their walking ability on a 5-mm narrow path (ledge test) and in an open field based on a structured scoring system ranging from 0 (no impairment) to 3 (maximal impairment) [38]. These evaluations were performed by trained investigators blind to the source (Pt-IgG vs. Ctrl-IgG) of the antibodies received by each mouse, with the final scores computed as the median values of the scores for each test [39]. With the same approach, investigators also scored hindlimb clasping, which is a marker of disease progression in a number of mouse models of neurodegeneration including certain cerebellar ataxias, and kyphosis, which is a common manifestation of neurodegenerative disease in mouse models [38].

### 2.10. Assessment of Licking Behavior

Water-licking behavior [40] was recorded as previously described in detail [39] during the 12-h dark period in the mouse home cage. Briefly, licking was recorded by measuring the junction potential that occurred between the steel spout of a water bottle and an aluminum foil on the cage bottom every time the mouse licked the spout standing on the foil, thereby closing the electrical circuit with its body [40]. The voltage signals were manually reviewed to reject artifacts. The inter-lick time interval was computed within trains of ≥4 licks and its median value over the whole recording was retained for each mouse [39].

### 2.11. Statistical Analysis

The statistical analysis was performed with SPSS (SPSS, Armonk, NY, USA). The ledge test scores at d14 and d45 were analyzed with Mann-Whitney U tests. The other results were analyzed with mixed-model analysis of variance (ANOVA) with general linear model procedure. The ANOVA within-subject factors for different analyses included time (2 levels: d14 vs. d45), the light-dark period (2 levels, light vs. dark), the sleep state (2 levels: NREM vs. REM sleep) and the NREM sleep bout duration bin (3 levels: 4–8 s, 12–120 s, >120 s, with Huyn-Feldt correction in case of failure of the sphericity assumption). The ANOVA between-subject factors were the mouse strain (2 levels: WT vs. hTau mice) and the source of the IgG received by each mouse (2 levels: Pt-IgG vs. Ctrl-IgG). Normality of the data distributions was evaluated with repeated Shapiro-Wilk tests (SPSS EXAMINE command) with false-discovery rate correction. Data were square-root or log-transformed when necessary to ensure normality.

The main analysis was performed on the female WT and hTau mice that completed both the d14 and the d45 phenotyping rounds. Ancillary analyses were performed on all the female WT and hTau mice that completed the 14d phenotyping round, also including the mice whose implants failed before d45. Replication analyses were performed on male hTau mice that completed both the d14 and the d45 phenotyping rounds. All tests were two-tailed, with statistical significance set at *p* < 0.05. The sample size and the detailed statistical results of each test are reported in Table A1, Table A2, Table A3, Table A4, Table A5, Table A6, Table A7, Table A8, Table A9, Table A10 and Table A11. Data are reported as means ± SEM. Graphic were plotted using R and its ggplot2 package.

## 3. Results

### 3.1. Specificity of the Purified Antibodies

The purified Pt-IgG showed the expected pattern of reactivity with IgLON5-espressing transfected HEK cells and with rat cerebellar and hippocampal tissue. Specific immunoabsorption with live HEK293 cells transfected to express human IgLON5 totally abrogated the reactivity, which was maintained instead after mock absorption with non-transfected cells (data not shown). Negative results were obtained with Ctrl-IgG (Appendix A).

### 3.2. Neuropathology

The neuropathological assessment blind to the Pt-IgG vs. Ctrl-IgG treatment revealed a mild, granular immunoreactivity in the cytoplasm of neurons of some mice, especially in the hippocampal CA4 region and mossy fibers and in the posterior sections around the ependyma (Figure 2). This immunoreactivity pattern varied between animals, so no distinctive brain nucleus or region could be identified. Discrimination of female WT and hTau mice based on higher vs. lower immunoreactivity by three blind investigators correctly identified Pt-IgG vs. Ctrl-IgG treated mice in 79% of cases (15/19 mice), consisting of 8/10 mice with Pt-IgG (3/3 WT and 5/7 hTau mice) and 7/9 mice with Ctrl-IgG (3/4 WT and 4/5 hTau mice). By contrast, identification was correct only in 1/5 vs. 1/2 male hTau mice with Pt-IgG vs. Ctrl-IgG.

### 3.3. Daily EMG Activity Profiles

Treatment with Pt-IgG did not significantly disrupt the daily EMG activity profiles of female WT and hTau mice at d14 and d45, which showed the expected increase during the dark (active) period compared to the light (rest) period (Figure 3 and Appendix A, Table A1 in Appendix B). Similar results were obtained in ancillary analyses on female WT and hTau mice at d14 including mice that did not complete experiments at d45, as well as in replication analyses on male hTau mice at d14 and d45 (Appendix A and Table A1).

### 3.4. Sleep Phenotype

During the first 8 h of the light (rest) period, female WT and hTau mice with ICV infusion of Pt-IgG did not show any significant difference in the time spent in wakefulness, NREM sleep, and REM sleep with respect to mice infused with Ctrl-IgG, spending most of the time in NREM sleep both at d14 and at d45, as expected (Figure 4 and Table A2, Table A3 and Table A4). Similar results were obtained in ancillary analyses on female WT and hTau mice at d14 only, including mice that did not complete experiments at d45, as well as in male hTau mice at d14 and d45 (Appendix A and Table A2, Table A3 and Table A4).

Inspection of raw EEG, EMG, and ventilation tracings during NREM sleep and REM sleep did not reveal obviously abnormal patterns in the mice that received Pt-IgG (Appendix A). The distribution of NREM sleep bout duration (Figure 5A,B, Table A5) and the neck EMG (Figure 5C,D, Table A6) and EEG activity (theta/delta power ratio; Figure 5E,F, Table A7) during sleep did not differ significantly between Pt-IgG and Ctrl-IgG in female WT and hTau mice at d14 and d45. Similar results were obtained in ancillary analyses on female WT and hTau mice at d14 only, including mice that did not complete experiments at d45, as well as in male hTau mice at d14 and d45 (Appendix A and Table A5, Table A6 and Table A7). As expected, all these analyses indicated significant effects of the sleep states, with higher EMG and lower theta/delta EEG ratio during NREM sleep than during REM sleep (Table A6 and Table A7). These analyses also indicated relatively robust significant main effects of the mouse strain and time on the theta/delta EEG ratio (Table A7).

### 3.5. Sleep-Related Breathing Phenotype

As expected [34], sleep apneas were detected during NREM sleep and REM sleep, and were often preceded by augmented breaths (sighs) in NREM sleep (Figure 6A). The occurrence rate of sleep apneas did not significantly depend on Pt-IgG vs. Ctrl-IgG administration in female WT and hTau mice at d14 and d45 (Figure 6B,C, Table A8). A decrease in the sample mean of apnea occurrence rate was observed during REM sleep in mice treated with Ctrl-IgG but was not supported statistically. Similar results were obtained in ancillary analyses on female WT and hTau mice at d14 only, including mice that did not complete experiments at d45, as well as in male hTau mice at d14 and d45 (Appendix A and Table A8).

A reproducible trend for an increase in ventilatory period during sleep was found in female WT and hTau mice treated with Pt-IgG vs. Ctrl-IgG at d14 and d45 (*p* = 0.011, Figure 6D,E and Table A9; grand means: 370 ms vs. 333 ms with Pt-IgG vs. Ctrl-IgG). Sample means were invariably higher with Pt-IgG than with Ctrl-IgG in female WT and hTau mice at d14 only, including mice that did not complete experiments at d45 (*p* = 0.237; Table A9; grand means: 360 ms vs. 338 ms with Pt-IgG vs. Ctrl-IgG) and in male hTau mice at d14 and d45 (*p* = 0.395; Appendix A and Table A9; grand means: 347 ms vs. 319 ms with Pt-IgG vs. Ctrl-IgG), although differences were not statistically significant.

### 3.6. Ataxia and Gait Instability

The motor function tests did not reveal any obvious functional alteration due to Pt-IgG in mice (Table A10). Among female WT and hTau mice, the ledge test showed at most a mild impairment (score 1 out of 3), which occurred in 2/11 vs. 4/9 mice with Pt-IgG vs. Ctrl-IgG at d14, and in 1/8 vs. 2/8 mice with Pt-IgG vs. Ctrl-IgG at d45, with no significant difference associated with anti-IgLON5 antibodies (*p* = 0.331 and *p* = 0.721 at d14 and d45, respectively). Similarly, in male hTau mice, a score of 1 at the ledge test was assigned to 1/5 vs. 1/3 mice with Pt-IgG vs. Ctrl-IgG at d14, and to 0/4 mice vs. 3/3 mice with Pt-IgG vs. Ctrl-IgG at d45 (*p* = 0.786 and *p* = 0.057 at d14 and d45, respectively). A mild impairment in walking in the open field (score 1 out of 3) and a mild hindlimb clasping (score 1 out of 3) were recorded only at d14 in one male hTau mouse with Pt-IgG, which also had a mildly impaired ledge test. Kyphosis was not observed in any mouse tested.

### 3.7. Licking Behavior

Inspection of raw tracings did not reveal any obvious alteration of licking behavior in mice with Pt-IgG (Figure 7A). Data in female WT and hTau mice at d14 and d45 indicated a reproducible trend for a decrease of the inter-lick interval associated with Pt-IgG (*p* = 0.015; Figure 7B,C and Table A11; grand means: 124 ms vs. 132 ms with Pt-IgG vs. Ctrl-IgG). Sample means were invariably lower with Pt-IgG than with Ctrl-IgG treatment in female WT and hTau mice at d14 only, including mice that did not complete experiments at d45 (*p* = 0.101, Table A11: grand means: 127 ms vs. 131 ms with Pt-IgG vs. Ctrl-IgG) and in male hTau mice at d14 and d45 (*p* = 0.948, Appendix A and Table A11; grand means: 133 ms vs. 134 ms with Pt-IgG vs. Ctrl-IgG), although differences were not statistically significant.

## 4. Discussion

We performed a pilot study of whether ICV infusion of mice with IgG from a patient with anti-IgLON5 disease (Pt-IgG) recapitulated the neuropathological and clinical features of the disease. Intriguingly, the brains of female WT and hTau mice that received Pt-IgG could be largely discriminated from those of the female mice that received Ctrl-IgG based on a greater extent of p-Tau deposition in the hippocampus and brainstem, as assessed by subjective evaluation of AT8 antibody IHC staining by three blind scorers. In female WT and hTau mice studied at the end of the 14-day antibody infusion period (d14) and, again, at least 30 days thereafter (d45), we found subtle but reproducible trends in the effects of Pt-IgG on ventilatory period during sleep (increased) and on the inter-lick interval during wakefulness (decreased), while sleep, respiratory, and motor control appeared otherwise preserved. However, we were unable to confirm these neuropathological and functional findings on a small group of male hTau mice studied at d14 and d45.

Although our pilot study did not yield conclusive evidence in support of a direct pathogenic role of Pt-IgG, it nevertheless yielded useful information to move forward. This information may be summarized as follows: (1) our preliminary results on enhanced p-Tau deposition at least in female hTau and WT mice infused with Pt-IgG warrant replication on WT mice of either sex for generalizability, with the full set of control experiments that were not included in our pilot study (see limitations section below); (2) our data were not consistent with major effects of Pt-IgG on sleep, respiratory, and motor control in mice, which could be expected based on the severity of the human disease: future work on mice should aim for subtler effects and set sample size accordingly; (3) future work may attempt to replicate our findings on ventilatory period during sleep and inter-lick interval during wakefulness, which were not anticipated based on the clinical features of human disease.

Multiple lines of evidence suggest that p-Tau deposits are a primary driver of neurodegeneration in Alzheimer disease [41], and this may be the case also for anti-IgLON5 disease. Nevertheless, caution is needed in interpreting the present results. We found, at most, evidence of a mild and diffuse tauopathy, and did not assess the occurrence of neurodegeneration. Some mice treated with Ctrl-IgG were erroneously attributed to the Pt-IgG group based on blind assessment of their AT8 staining, suggesting adverse effects of the chronically indwelling bilateral ICV cannula, non-specific adverse effects of human IgG infusion, or experimental variability in IHC staining. In this respect, p-Tau deposition may be detected with AT8 antibodies also in mouse models of epilepsy [42] and diabetes [43]. The fraction of female WT mice that were correctly classified as treated with Pt-IgG vs. Ctrl-IgG based on their AT8 IHC staining was even higher than that of female hTau mice. This finding may suggest that in female hTau mice, any enhancement of Pt-IgG effects due to the increased strain susceptibility to p-Tau deposition was offset by increased inter-individual variability, perhaps due to interindividual differences in *MAPT* transgene expression. Substantial differences were, indeed, observed in AT8 staining between the brains of two intact hTau founder mice (data not shown). In addition, social isolation, which was meant to prevent damage from conspecifics after instrumentation (see Methods), may have contributed to increase Tau protein phosphorylation also in WT mice, as it has been shown to do in middle-aged WT rats [44].

The total sample size of our pilot study was substantial given the complexity of the experimental procedures, yet our study was arguably underpowered to detect minor phenotypic differences. Nevertheless, our results can help design future studies with sufficient statistical power and suggest that chronic ICV infusion of Pt-IgG in mice may not recapitulate the dramatic clinical picture of the human anti-IgLON5 disease, at least up to more than 45 days after the start of the infusion.

Minor alterations in ventilatory period during sleep and in inter-lick interval during wakefulness due to Pt-IgG were suggested, but not robustly demonstrated by our pilot study due to lack of internal replication. Neither of these alterations was expected based on the available clinical evidence on human anti-IgLON5 disease. Nevertheless, these results warrant replication on larger samples, as they may contribute to shed light on IgLON5 disease pathophysiology. Indeed, the unpredictability of the results of administration of human antibodies to experimental animal models has long been remarked [19]. An increase in ventilatory period, corresponding to a decrease in breathing rate, might reflect an increase in airway resistance [45], consistent with the reported occurrence of stridor during sleep in patients with anti-IgLON5 disease [5]. An increase in inter-lick interval would have been expected as a result of Pt-IgG, mirroring bulbar motor dysfunction in patents with IgLON5 disease [1]. On the other hand, the observed decrease in inter-lick interval with Pt-IgG may reflect mild behavioral stress during wakefulness [46].

We were unable to replicate on male hTau mice the findings on p-Tau deposition, ventilatory period during sleep, and inter-lick interval during wakefulness, which we obtained on female WT and hTau mice. This discrepancy may have resulted from false-positive results on female WT and hTau mice, false-negative results on our small group of male hTau mice, or true sex-related differences. Since we did not study male WT mice, we are unable to suggest whether this sex difference may be generalized or limited to hTau mice, perhaps due to greater inter-individual variability in p-Tau deposition. Although no gender difference in human anti-IgLON5 disease is apparent, any protective or adverse pathophysiological effects of sex hormones might be obscured in human patients due to the late onset of the disease [1]. Our data, thus, warrant replication on male and female WT mice with sufficient sample size to detect any potential sex-related difference in young adult individuals of this species.

We cannot exclude the contribution of other technical details of this pilot study design to the variability of the results. The mice we studied may have been too young to develop full-blown disease. In particular, we studied relatively young mice at 19–22 weeks of age in order to avoid a ceiling effect on p-Tau deposition in older hTau mice irrespective of Pt-IgG infusion [23,24,29], whereas human anti-IgLON5 disease typically occurs in older adults and in the elderly [1]. In addition, mice may not be susceptible to anti-IgLON5 antibodies due to species differences vs. human subjects. However, mice do express IgLON5 in the brain, particularly in the thalamus, pons, medulla oblongata, ventral striatum, and olfactory bulb [4], the IgLON5 sequence is highly conserved between mice, rats, and human subjects, and Pt-IgG react with rat brain tissue (Appendix A). The C57Bl/6J mouse strain might be relatively protected from adverse effects of anti-IgLON5 antibodies. Accordingly, one study suggested that the C57BL/6J genetic background weakens the phenotype of hTau mice [47]. The duration (14 days) and flow rate (0.5 μL/h) of ICV infusion and the antibody concentration (2 mg/dL) previously proved effective for ICV infusion of mice with human anti-NMDA receptor antibodies [28], but may be insufficient with anti-IgLON5 antibodies. A ≥ 45-day lag time between start of infusion and phenotypic and neuropathological evaluation may be insufficient to elicit full-blown disease. However, substantially increasing this lag may be technically challenging. Chronic ICV infusion of mice with Pt-IgG proved feasible, but the stability of the head implant including ICV cannulas and electrodes was a limiting factor for long-term studies. Since our study did not detect major significant alterations in EEG and EMG due to Pt-IgG, future experiments could be designed with simpler, lighter, and more stable head implants limited to ICV cannulas to increase the duration of the head implants. Alterations in sleep and sleep-related breathing, which are hallmarks of the human anti-IgLON5 disease [2,5], may be accurately phenotyped in freely-behaving mice within a whole-body plethysmograph even without EEG and EMG recordings [36].

Our pilot study had several limitations, which should be addressed in future larger-scale experimentation. Our study was framed as a pilot study covering many approaches to provide us and the research community with useful advice. As a result, while the total sample size of our study was quite substantial, the sample size per group was limited, with the potential for false-negative statistical results. On the other hand, the analyses were specifically designed to minimize risk of false-positive results, by relying on ANOVAs instead of simple comparisons, and by explicitly differentiating between main analyses and confirmatory ancillary/replication analyses. Applying a family-wise significance correction would have been overly conservative for a pilot study. Nevertheless, we acknowledge that given the extensive phenotyping of multiple variables that we performed, a risk of false-positive results represented a study limitation.

As previously discussed, we designed our pilot study focusing on female hTau mice, surmising that the propensity of hTau mice to p-Tau brain accumulation would boost effects of Pt-IgG, and taking account of the available evidence of greater brainstem p-Tau deposition in female than in male hTau mice [25]. We included a control group of female C57Bl/6J WT mice to test the generalizability of data on female hTau mice, also considering that C57Bl/6J mice are arguably the most widely studied inbred mouse strain. Since contrasting results on p-Tau deposition in female vs. male hTau mice were subsequently published [26,27], we included a further replication group of male hTau mice while the study was ongoing. While this represented a point of strength of our study, also considering the lack of gender-specificity of human anti-IgLON5 disease, the lack of a further male WT control group represented a study limitation, particularly in light of our negative findings on male hTau mice. Our study was not designed to address the issue of sex-dependent differences in p-Tau deposition in hTau mice, which would have requires studying male and female hTau mice of multiple age groups infused with Ctrl-IgG.

In our pilot study, we euthanized mice for brain harvesting a few weeks after the end of the antibody infusion period to account for possible delayed effects. However, the processes of antibody washout from cerebrospinal fluid and the clearance from brain tissue that occurred between the end of the infusion and the tissue fixation prevented our ability to correlate the phenotypic and neuropathological changes in individual mice with the effective extent of IgG penetration into the brain tissue. This represents a limitation of our work. Accordingly, no IgG deposits were detected in the brain sections of the mice under study (data not shown). In this respect, future studies may evaluate the kinetics of IgG clearance from the brain after the end of ICV antibody infusion.

Another limitation of our pilot study is that p-Tau deposition was evaluated only based on AT8 IHC staining. Our results should be replicated using a range of antibodies targeting different p-Tau epitopes. Moreover, although the subjectivity in our procedures was mitigated because the analysis was performed by three blind researchers, our results should be replicated also with objective measurements of p-Tau deposition, such as by Western Blot. Future studies should also test whether Pt-IgG cause internalization of IgLON5 in vivo, as it has been shown to occur in vitro [20], and address neuroinflammatory changes, which may be involved in the pathophysiology of autoimmune neurological disorders.

We employed antibodies that were carefully tested and purified from one patient with typical clinical features and from one control subject. While this was meant to limit variability in the data, the lack of replication groups with antibodies purified by other patients represents yet another limitation of our pilot study.

## 5. Conclusions

The results of our pilot study suggest, but do not prove, that chronic ICV infusion of mice with IgG from a patient with anti-IgLON5 disease may elicit neuropathological, respiratory, and motor alterations. Considering the preliminary nature of our pilot study and acknowledging the difficulty of recapitulating clinical manifestations of a heterogeneous disease such as anti-IgLON5 disease, these results warrant replication and clarification on larger samples also taking account of potential sex-related differences. This research has the potential to shed light on the pathophysiology of this enigmatic disease at the borderland between autoimmunity and neurodegeneration.

## Figures and Tables

**Figure 1 cells-11-01024-f001:**
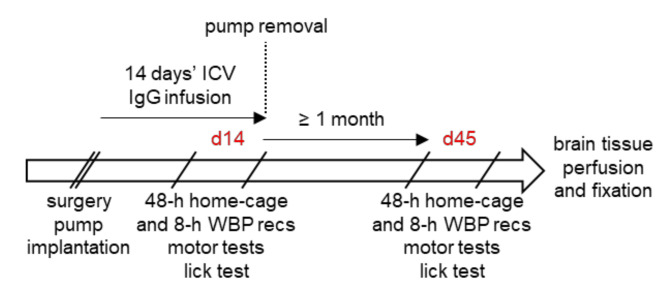
Schematic of the experimental protocol. ICV: intracerebroventricular. IgG: G immunoglobulin. WBP: whole-body plethysmograph. Recs: recordings. The two phenotyping rounds were performed at the end of the 14-day ICV antibody infusion (d14) and at least 1 month after the end of the infusion (d45). Surgery and pump implantation was performed in mice at 19–22 weeks of age.

**Figure 2 cells-11-01024-f002:**
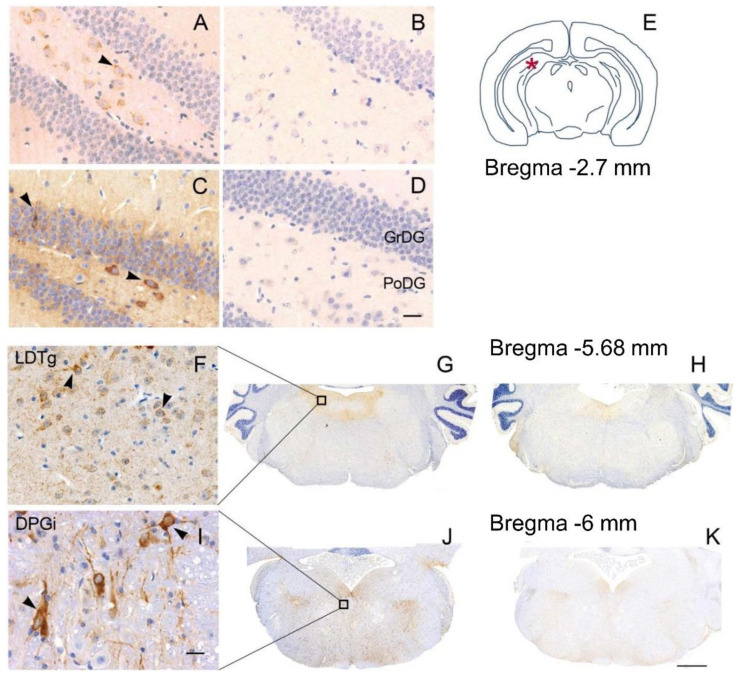
Representative sections from hippocampus (**A**–**D**) and brainstem (**F**–**K**) of wild type mice (**A**,**B**,**F**–**H**) and female hTau mice (**C**,**D**,**I**–**K**) infused with IgG from a patient with anti-IgLON5 disease (**A**,**C**,**F**,**G**,**I**,**J**) or from a control subject (**B**,**D**,**H**,**K**). The asterisk in (**E**) indicates the area of analysis of the phosphorylated Tau protein deposition in the hippocampus at the reported bregma coordinate. (**F**,**I**) are magnifications of the areas indicated by the squares in (**G**,**J**), respectively. The pathology appeared more severe in hTau than in wild-type mice. Scale bars: (**D**,**I**) = 20 μm; (**K**) = 500 μm. PoDG: polymorph layer of the dentate gyrus; GrDG: granule cell layer of the dentate gyrus; LDTg: laterodorsal tegmental nucleus; DPGi: dorsal paragigantocellular nucleus.

**Figure 3 cells-11-01024-f003:**
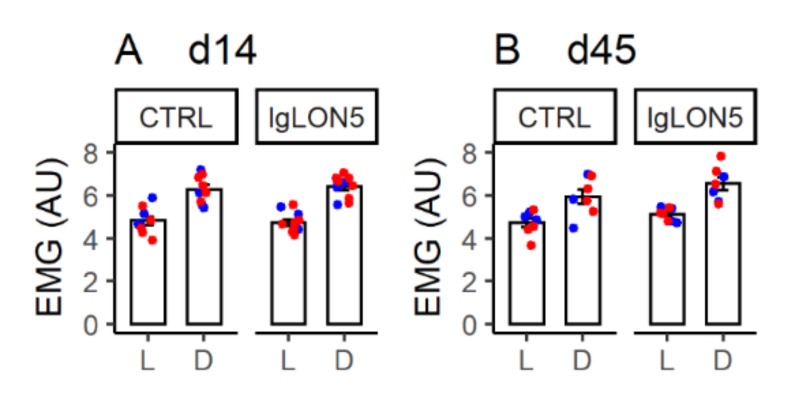
Average neck muscle electromyogram root mean square (EMG, in arbitrary units, AU) during the light (L) and dark (D) period in female wild-type (blue) and hTau (red) mice with intracerebroventricular infusion of antibodies from a patient with anti-IgLON5 disease (IgLON5) or from a control (CTRL) subject. Panels (**A**,**B**) show results obtained at the end of the 14-day infusion (d14) and ≥1 month afterward (d45). Bars show means ± SEM. Dots show values in individual mice. Detailed statistical results are reported in Table A1.

**Figure 4 cells-11-01024-f004:**
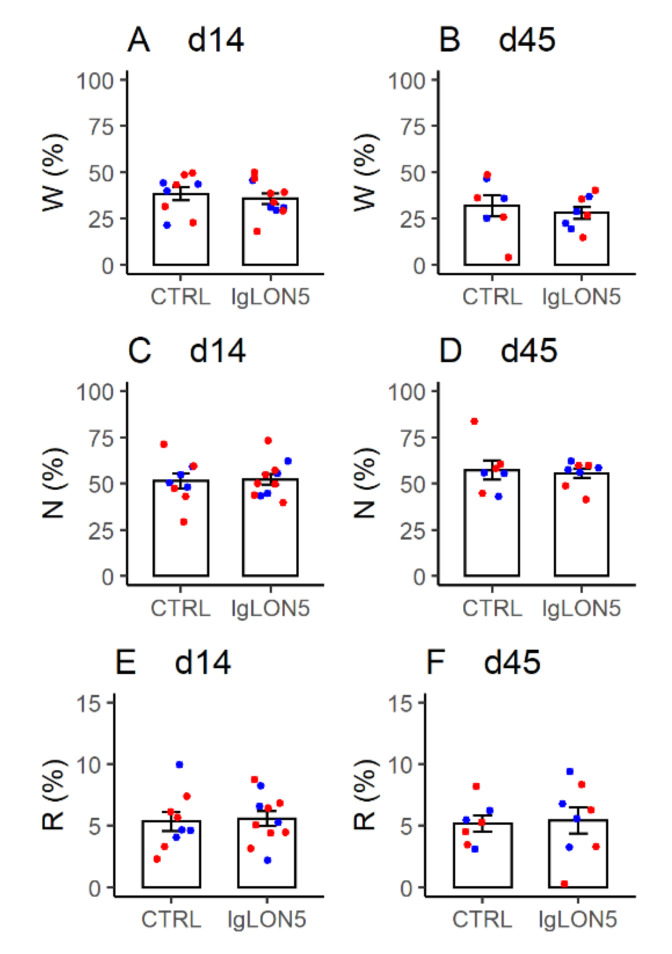
Percentage of the recording time spent in wakefulness (W), NREM sleep (N), and REM sleep (R) during the first 8 h of the light period by female wild-type (blue) and hTau (red) mice with intracerebroventricular infusion of antibodies from a patient with anti-IgLON5 disease (IgLON5) or from a control (CTRL) subject. (**A**,**C**,**E**): results obtained at the end of the 14-day infusion (d14). (**B**,**D**,**F**): results obtained ≥ 1 month afterward (d45). Bars show means ± SEM. Dots show values in individual mice. Detailed statistical results are reported in Table A2, Table A3 and Table A4.

**Figure 5 cells-11-01024-f005:**
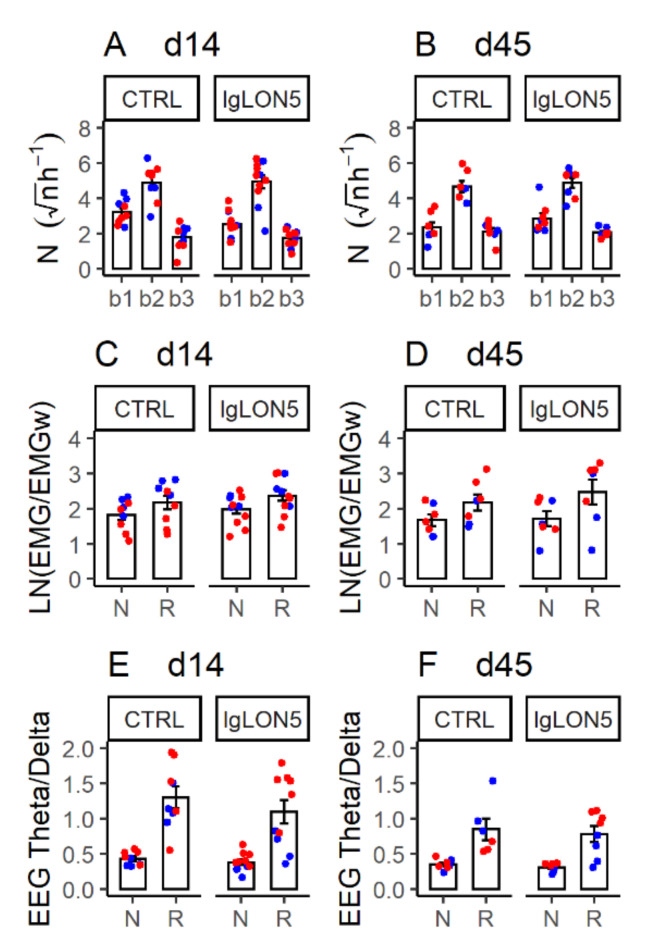
Indexes of sleep architecture (**A**,**B**) and quality (**C**,**F**) in female wild-type (blue) and hTau (red) mice with intracerebroventricular infusion of antibodies from a patient with anti-IgLON5 disease (IgLON5) or from a control (CTRL) subject. (**A**,**B**): square-root transformed numbers of episodes of NREM sleep with duration 4–8 s (b1), 12–120 s (b2) or >120 s (b3) per hour recording time during the first 8 h of the light period. (**C**,**D**): log-transformed ratios of electromyogram (EMG) root mean square during sleep (N: NREM sleep; R: REM sleep) and during wakefulness (EMGw). (**E**,**F**): ratios of electroencephalogram (EEG) theta and delta power during N and R. (**A**,**C**,**E**): data obtained at the end of the 14-day infusion (d14). (**B**,**D**,**F**): results obtained ≥ 1 month afterward (d45). Bars show means ± SEM. Dots show values in individual mice. Detailed statistical results are reported in Table A5, Table A6 and Table A7.

**Figure 6 cells-11-01024-f006:**
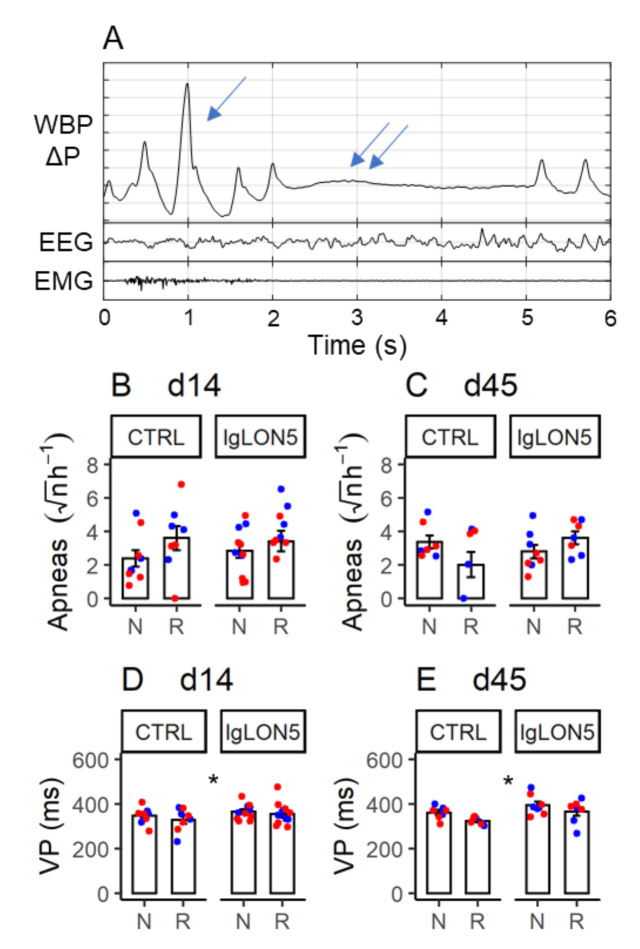
(**A**): representative raw tracings including an augmented breath (sigh, arrow) and a missed breath (apnea, double arrow) during NREM sleep in a female hTau mouse upon termination of the 14-day period of intracerebroventricular infusion with antibodies from a patient with anti-IgLON5 disease. WBP ΔP: pressure difference between whole-body plethysmograph chambers, representing the breathing signal (inspiration upwards). EEG: electroencephalogram. EMG: electromyogram. (**B**,**C**): square-root transformed number of apneas per hour of NREM (N) or REM (R) sleep time. (**D**,**E**): ventilatory period (VP) during N and R. (**B–E**): data from female wild-type (blue) and hTau (red) mice infused with antibodies from patients with anti-IgLON5 disease (IgLON5) or from control (CTRL) subjects. (**B**,**D**): data obtained at the end of the 14-day infusion (d14). (**C**,**E**): results obtained ≥ 1 month afterward (d45). Bars show means ± SEM. Dots show values in individual mice. * *p* < 0.05, ANOVA main effect of IgLON5 vs. CTRL antibodies. Detailed statistical results are reported in Table A8 and Table A9.

**Figure 7 cells-11-01024-f007:**
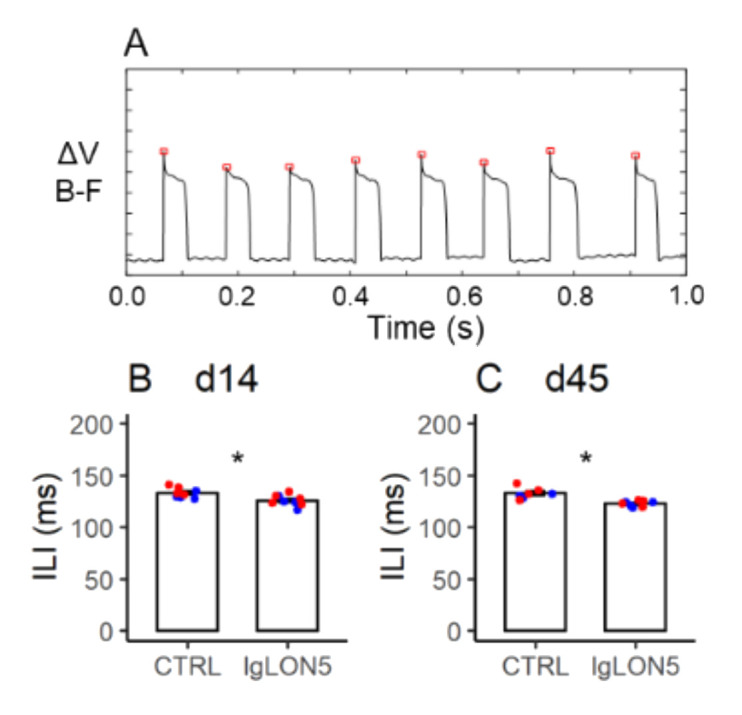
Panel (**A**) shows representative raw tracings of licking behavior in a female hTau mouse upon termination of the 14-day period of intracerebroventricular infusion with antibodies from a patient with anti-IgLON5 disease. ΔV (**B**–**F**): electric potential difference between the steel nozzle of the water bottle and an aluminum plate on the cage floor beneath the mouse. Upward deflections of ΔV (**B**–**F**) correspond to individual licks. The inter-lick intervals (ILI) are measured as the time intervals between the peaks of ΔV (**B**–**F**) deflections (red squares). (**B**,**C**): values of ILI at the end of the 14-day infusion (d14) and ≥ 1 month afterward (d45) in female wild-type (blue) and hTau (red) mice infused with antibodies from patients with anti-IgLON5 disease (IgLON5) or from control (CTRL) subjects. Bars show means ± SEM. Dots show values in individual mice. * *p* < 0.05, ANOVA main effect of IgLON5 vs. CTRL antibodies. Detailed statistical results are reported in Table A11.

## Data Availability

The data presented in this study are available from the corresponding author upon reasonable request.

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
