# Peer review of "Pilot Study of the Effects of Chronic Intracerebroventricular Infusion of Human Anti-IgLON5 Disease Antibodies in Mice"

_cells, 2022, doi:10.3390/cells11061024_

Round 1

Reviewer 1 Report

In order to study the effects of anti-IgLON5 antibodies on behavior and on P-Tau accumulation, the authors infused polyclonal IgG from patients with IgLON5 disease in mice, either WT or expressing the human Tau-protein (hTau). They find a delayed (6 weeks) increase in augmented breath events and decrease of the normal licking behavior in treated female mice, regardless of the hTau status. There was no effect on male hTau mice (WT males were not included).

These results are very limited, suggesting Iglon5 antibodies have no effect in the animals. However, there is not enough information to know whether this negative result is in fact due to technical limitations.

Major comments:

Because they are so many subgroups, the resulting sample sizes are small, which may explain the negative results. In addition, hTau does not seem to have any impact on the parameters studied here. Why the authors did not go on with their study, focusing on female WT mice? Likely the results would be easier to analyze.

The authors should clarify why they used hTau but not WT male mice.

The authors show that P-IgGs target IgLON5, but additional antigenic targets were not ruled out. This should be done with an immunoabsorption experiment, or by testing Pt-IgG on the brain of an IgLON5 KO mouse.

The authors should demonstrate that the protocol of antibody infusion was performed correctly, by demonstrating the presence of human IgG deposits in the mice brains.

The estimation of Tau expression is based on subjective analysis. The authors should use instead a validated quantitative or semi-quantitative method, for instance western blot analysis.

The study design is relatively complex; the authors studied many parameters in multiple groups of animals. How did they adjust for such multiple testing?

In cultured neurons, IgLON5 antibodies induce an irreversible internalization of IgLON5. It is necessary to quantify IgLON5 expression in the brain of the infused mice in order to assess if there was an effect of the antibodies at least at the molecular level.

IgLON5-IgG are mostly IgG1 and/or IgG4, and because these subclasses have different properties, their effect could be different. What were the respective proportions of IgG1/IgG4 subclasses in the IgG pool used in this study? What is more, Iglon5 disease is clinically heterogeneous. How many patients were used to form the IgG pool? What was the phenotype (bulbar, cerebellar, sleep disorder…) of the patients included? This as well could influence the results.

There is seemingly less Tau accumulation in male mice. The authors explain that it may be because in this mouse strain the accumulation of P-Tau is slower in males. Then, why WT female mice have some effect?

The authors explain that augmented breath events may be an indirect sign of apnea, (although apnea events per se were not increased in treated animals). Are there any reference supporting this in the literature?

Minor comments:

The authors write twice “there was a difference in (…) at day 14 and day 45. The significance of this was not confirmed at day 14.”  The correct sentence would be: there was a significant difference at day 45 and not at day 14. (alternative: there was only a trend at day 14)

“Mice immunized with antibodies” (line 348) is not correct. It suggests the mice developed an immune response against human antibodies. It should be instead “injected” or “infused” with antibodies.

Author Response

We thank the Reviewer for thoughtful comments and criticism. We carefully and extensively revised our manuscript in line with the Reviewer’s requests and suggestions. Please find below our point-by-point responses. 

COMMENT #R1.1:

These results are very limited, suggesting Iglon5 antibodies have no effect in the animals. However, there is not enough information to know whether this negative result is in fact due to technical limitations.

RESPONSE:

We acknowledged more extensively the limitations of our study in the revised Discussion section. Many of these limitations are consistent with the fact that our study was meant as a preliminary pilot study to guide larger-scale experimentation, as we made clear in the study title and throughout the manuscript. Nevertheless, we contend that our study did produce useful information, which may be summarized as follows: 1) our results were suggestive of greater p-Tau deposition in the brainstem and hippocampus at least in female hTau and WT mice infused with anti-Iglon5 antibodies: future work should attempt to replicate this finding on female and male WT mice; 2) our data were not consistent with any major effect of anti-IgLON5 antibodies on sleep, respiratory, and motor control in mice: future work should aim for subtler effects and set sample size accordingly; 3) we found reproducible trends in ventilatory period during sleep and inter-lick interval during wakefulness associated with anti-IgLON5 antibody infusion, which were not anticipated based on the clinical features of human disease: future work on mice may attempt to replicate these unexpected findings. These considerations are now clearly addressed in the revised Discussion section.   

Major comments:

COMMENT #R1.2:

Because they are so many subgroups, the resulting sample sizes are small, which may explain the negative results. In addition, hTau does not seem to have any impact on the parameters studied here. Why the authors did not go on with their study, focusing on female WT mice? Likely the results would be easier to analyze.

RESPONSE:

We acknowledge the limitation highlighted by the Reviewer. Our study was framed as a pilot study covering many approaches to provide us and the research community with useful advice. As a result, while the total sample size of our study was quite substantial, the sample size per group was limited. Nevertheless, our data were clearly not consistent with major effects of anti-IgLON5 antibodies on sleep, respiratory, and motor control in mice, which could be expected based on the severity of the human disease. It is in light of the results of our pilot study that we now recommend performing future work on this topic on female and male WT mice, rather than on hTau mice with a genetic propensity to p-Tau deposition. These considerations are summarized in the revised Discussion section.

COMMENT #R1.3

The authors should clarify why they used hTau but not WT male mice.

RESPONSE:

As detailed in the Introduction section of the original manuscript, the lack of a group of male WT mice was because our study was initially designed to focus on female hTau mice. At the time of study design, we focused on female hTau mice because evidence indicated that brainstem p-Tau deposition was greater in female than in male hTau mice already at 4 months of age. We included a control group of female WT mice to test the generalizability of data on female hTau mice. However, contrasting results on sex differences in p-Tau deposition in hTau mice were subsequently published, and we included a further replication group of male hTau mice while the study was ongoing. We think that this represented a point of strength of our study, also considering the lack of gender-specificity of human anti-IgLON5 disease. Nevertheless, we acknowledge that the lack of a male WT mice group in our pilot study is a limitation of our work. These considerations are more clearly reported in the revised Discussion section.

COMMENT #R1.4

The authors show that P-IgGs target IgLON5, but additional antigenic targets were not ruled out. This should be done with an immunoabsorption experiment, or by testing Pt-IgG on the brain of an IgLON5 KO mouse.

RESPONSE:

The IgGs were tested in vitro not to include extra reactivities by immunoabsorption before injecting the animals. This is now clarified in the revised Results section. The revised Methods section is modified accordingly. In particular, the binding of purified Pt-IgG and Ctrl-IgG was assessed with transfected HEK cells expressing human IgLON5 and histologically with rat tissue. To confirm the specificity of Pt-IgG, the antibodies were immunoabsorbed with live HEK293 cells expressing IgLON5 or with mock transfected cells. After six rounds of immunoabsorption, the antibodies were applied to sections of rat brain and the reactivity developed as described. Specific immunoabsorption with IgLON5 totally abrogated the reactivity meanwhile was maintained after mock absorption.

COMMENT #R1.5

The authors should demonstrate that the protocol of antibody infusion was performed correctly, by demonstrating the presence of human IgG deposits in the mice brains.

RESPONSE:

In response to this comment, we checked for the presence of IgG deposits on the brain sections of the mice under study, and the result was negative. This was expected, because we euthanized mice for brain harvesting very long after the end of the antibody infusion period to account for possible delayed effects of the antibodies. In particular, brain tissue was obtained approximately 3 weeks after the end of antibody infusion from the 5 mice in which experiments were interrupted due to partial detachment of the head implant, and at least one month after the end of antibody infusion in the remaining mice. Nevertheless, this is now highlighted as a limitation of our pilot study, together with the suggestion to explore the kinetics of IgG clearance from the brain after the end of intracerebroventricular infusion in future experiments.

COMMENT #R1.6

The estimation of Tau expression is based on subjective analysis. The authors should use instead a validated quantitative or semi-quantitative method, for instance western blot analysis.

RESPONSE:

This point is now highlighted in the revised Discussion section as a limitation of our pilot study and as a recommendation for future work. We are unable to perform Western blot analysis on the brains of the mice we studied because we have not stored unfixed brain tissue. On the other hand, subjectivity in the evaluation of brain sections in our pilot study was mitigated by the fact that the analysis was performed by three blind researchers. This is now stressed in the revised Methods and Discussion sections.

COMMENT #R1.7

The study design is relatively complex; the authors studied many parameters in multiple groups of animals. How did they adjust for such multiple testing?

RESPONSE:

As remarked above, our study was framed as a pilot study covering many approaches to provide us and the research community with useful advice. The analyses were specifically designed to minimize risk of false-positive results, by relying on ANOVAs instead of simple comparisons, and by explicitly differentiating between main analyses (d14 and d45, female) and ancillary and replication analyses, which were meant to be confirmatory. Nevertheless, we acknowledge that given the extensive phenotyping of multiple variables that we performed, a risk of false-positive results remains. On the other hand, applying a family-wise significance correction for the different variables tested would have been overly conservative for a pilot study. These considerations are now highlighted among the study limitations in the revised Discussion section.

COMMENT #R1.8

In cultured neurons, IgLON5 antibodies induce an irreversible internalization of IgLON5. It is necessary to quantify IgLON5 expression in the brain of the infused mice in order to assess if there was an effect of the antibodies at least at the molecular level.

RESPONSE:

This important point is now also highlighted in the revised Discussion section as a limitation of our pilot study and as a recommendation for future work. Unfortunately, we are unable to perform Western blot analysis on the brains of the mice we included in our pilot study because we have not stored unfixed brain tissue.

COMMENT #R1.9

IgLON5-IgG are mostly IgG1 and/or IgG4, and because these subclasses have different properties, their effect could be different. What were the respective proportions of IgG1/IgG4 subclasses in the IgG pool used in this study? What is more, Iglon5 disease is clinically heterogeneous. How many patients were used to form the IgG pool? What was the phenotype (bulbar, cerebellar, sleep disorder…) of the patients included? This as well could influence the results.

RESPONSE:

The patient IgGs that were infused to the mice in our pilot study were purified from a single patient, a female aged 65 years. The patient was very typical clinically, with predominant bulbar phenotype and chronic evolution, with positivity to HLA susceptibility alleles and IgG4 predominance. The control IgGs were purified from a control subject matched for age and gender. This is now clarified in revised Methods section. The possibility that patient features influence the results and the opportunity to replicate results with IgG from different patients are also now mentioned in the revised Discussion section.

COMMENT #R1.10

There is seemingly less Tau accumulation in male mice. The authors explain that it may be because in this mouse strain the accumulation of P-Tau is slower in males. Then, why WT female mice have some effect?

RESPONSE:

Published evidence on sex-related differences in p-Tau accumulation in hTau mice is contrasting. Our study was not designed to address this issue, which would have requires studying male and female hTau mice infused with control IgG at different ages. Our inability to correctly classify male hTau mice infused with patient vs control IgG based on the immunohistochemical pattern of p-Tau deposition may have resulted from increased interindividual variability of p-Tau deposition in male vs female hTau mice and/or from a genuine protective effect of male sex in mice. These considerations are now better clarified in the revised Discussion section.

COMMENT #R1.11

The authors explain that augmented breath events may be an indirect sign of apnea, (although apnea events per se were not increased in treated animals). Are there any reference supporting this in the literature?

RESPONSE:

Although augmented breaths (“sighs”) are often followed by apneas in mice, apneas do occur also irrespective of augmented breaths (cf. e.g., J Sleep Res. 2019 Dec;28(6):e12845). Therefore, we did not analyze augmented breaths in this manuscript, absent a direct translational relevance for patients with IgLON5 disease. This is now clarified in the revised Methods section.

Minor comments:

COMMENT #R1.12

The authors write twice “there was a difference in (…) at day 14 and day 45. The significance of this was not confirmed at day 14.”  The correct sentence would be: there was a significant difference at day 45 and not at day 14. (alternative: there was only a trend at day 14)

RESPONSE:

The statements referred to the ANOVA main effect of the anti-IgLON5 antibodies, which is evaluated statistically by taking account of data at d14 and d45. The absence of significant antibody (IgLON5 vs control) x time (d14 vs d45) interactions did not support the conclusion that significance was limited to results at d45. On the other hand, in the effort to fully report the available information, we reported also ancillary analyses of data at d14 only, which also included whose implants failed before d45. To summarize, while we contend that our reporting of results was statistically correct, this comment emphasizes that it was unclear. Nevertheless, in line with comment #11 of Reviewer 3, we reworded the Abstract, Results, and Discussion sections to acknowledge more simply the observed reproducible trends without overstating the results.

COMMENT #R1.13:

“Mice immunized with antibodies” (line 348) is not correct. It suggests the mice developed an immune response against human antibodies. It should be instead “injected” or “infused” with antibodies.

RESPONSE:

We reworded that expression throughout the revised manuscript, as suggested, and removed it from the keywords.

Reviewer 2 Report

28 January 2022

Review on the manuscript titled “Passive immunization with human anti-IgLON5 disease antibodies in mice: a pilot study” by Alvente S et al., submitted to Cells

Manuscript ID: cells-1590917

Dear Authors,

Anti-IgLON5 disease is a rare neurodegenerative autoimmune disease, characterized by parasomnias and chorea, associated with antibodies against a cell-adhesion protein IgLON5, and related to the accumulation of phosphorylated Tau protein in the brain stem. The authors studied the clinical and pathological consequence of passive intracerebroventricular immunization of antibodies from patients with Anti-IgLON5 disease in mice. The results showed decreases in breathing rate during sleep and interlick interval during wakefulness and expression of phosphorylated Tau proteins in female wild type and female human Tau protein expressing transgenic mice, but not in male mice of both types. The authors concluded larger studies are necessary to reveal sex differences in mice.

Please consider the following:

  1. Page 1, Title: Please include “Pilot study” in the title.
  2. A graphical abstract summarizing the manuscript is highly recommended.
  3. Page 1, Abstract:
  4. Please proportionally present background, purpose, methods, results, and conclusion.
  5. Page 1, Keywords: animal study or translation study is missing.
  6. Pages 1-2, Introduction:
    1. “HLA DRB1*10:01 and DQB1*05:01 alleles“, “immunotherapy”: Please describe a bit more about these alleles and immunotherapy.
    2. [14]: Probably the citation is not appropriate or rephrase the passage.
    3. Please briefly present autoimmune neurological diseases in general, including the role of inflammation in pathogenesis. Suggested references: https://doi.org/10.3390/diagnostics12010130; https://doi.org/10.3390/biomedicines10010076; https://doi.org/10.3390/biomedicines9070734; and doi: 10.1016/j.neubiorev.2020.08.010
    4. If there is no previous in vivo study, please clearly state this is the first study. If there is any, please describe as a part of background.
    5. Please present a rationale to use both wild type and transgenic mice.
  7. Pages 4-11, Methods, Results:
  8. Please present items of methods and results in the same order.
  9. ”Specificity of the purified antibodies”: Is this method presented?
  10. Please present p-values, asterisk in graphs, and statistical values in tables.
  11. Pages 6,11, Figures: Please use pointers.
  12. Pages 4-7, Discussion: Please discuss the previous studies and present study, weaknesses or limitation in the present study, potentials and significance of this study, the ultimate goal, research or knowledge needed to achieve, the future research direction, and the biggest challenge in this goal, among others.
  13. Pages 14-16, References: I recommend authors to use more evidence to back their claims, preferably more than 50 references for original articles. Thus, I recommend the authors to attempt to deepen the subject of their manuscript, as the bibliography is too concise: nonetheless, in my opinion, less than 50 articles for a research paper are really insufficient. Currently authors cite only 39 papers, and they are few. Therefore, I suggest the authors to focus their efforts on researching relevant literature: I believe that adding more studies will help to provide better and more accurate background to this study.

The manuscript contains eight figures, no table and 39 references. The manuscript carries important value trying to present in vivo evidence of contribution of sex factor in the pathogenesis of Anti-IgLON5 disease. I recommend this manuscript for publication after major revision.

Author Response

We thank the Reviewer for thoughtful comments and criticism. We carefully and extensively revised our manuscript in line with the Reviewer’s requests and suggestions. Please find below our point-by-point responses to the Reviewer’s comments.

COMMENT #R2.1

Page 1, Title: Please include “Pilot study” in the title.

RESPONSE:

We reworded the title to include “Pilot study” and remove the colon, hoping that this corresponds to the Reviewer’s request.

COMMENT #R2.2

A graphical abstract summarizing the manuscript is highly recommended.

RESPONSE:

We added a graphical abstract to the revised manuscript.

COMMENT #R2.3

Page 1, Abstract: Please proportionally present background, purpose, methods, results, and conclusion.

RESPONSE:

The revised manuscript now includes a structured abstract with a proportional representation of the sections mentioned by the Reviewer, including a shorter background section.

COMMENT #R2.4

Page 1, Keywords: animal study or translation study is missing.

RESPONSE:

The revised manuscript now includes the keyword “Animal models”, which is a PubMed MEsH term in line with the Reviewer’s suggestions.

COMMENT #R2.5

Pages 1-2, Introduction: “HLA DRB1*10:01 and DQB1*05:01 alleles“, “immunotherapy”: Please describe a bit more about these alleles and immunotherapy.

RESPONSE:

The revised Introduction section now clarifies that the association with these HLA alleles and the clinical response to immunotherapy support an autoimmune pathogenesis of anti-IgLON5 disease even though HLA allele positivity is not associated with a better response to immunotherapy.

COMMENT #R2.6

[14]: Probably the citation is not appropriate or rephrase the passage.

RESPONSE:

We confirm that the citation was appropriate. The text has been rephrased to make it clear.

COMMENT #R2.7

Please briefly present autoimmune neurological diseases in general, including the role of inflammation in pathogenesis. Suggested references: https://doi.org/10.3390/diagnostics12010130; https://doi.org/10.3390/biomedicines10010076; https://doi.org/10.3390/biomedicines9070734; and doi: 10.1016/j.neubiorev.2020.08.010

RESPONSE:

We revised the Introduction as suggested, also referring to the references proposed by the Reviewer.

COMMENT #R2.8

If there is no previous in vivo study, please clearly state this is the first study. If there is any, please describe as a part of background.

RESPONSE:

To the best of our knowledge, no in vivo studies similar to ours have been published previously. This is now remarked in the revised Introduction and Discussion sections.

COMMENT #R2.9

Please present a rationale to use both wild type and transgenic mice.

RESPONSE:

We studied hTau mice because they show a spontaneous tendency to p-Tau deposition and late-life neurodegeneration. We studied also C57Bl/6J wild-type mice to evaluate the generalizability of the results obtained in hTau mice, also considering that the C57Bl/6J strain is arguably the most widely studied inbred mouse strain. These considerations are now included in the revised Introduction section and better clarified in the revised Discussion section.

COMMENT #R2.10

Pages 4-11, Methods, Results: Please present items of methods and results in the same order.

RESPONSE:

We restructured the revised Methods section to present items in the same order as in the Results section, as suggested. Please note that in the revised manuscript, we presented results concerning neuropathology first, as suggested in comment #12 of Reviewer 3.

COMMENT #R2.11

“Specificity of the purified antibodies”: Is this method presented?

RESPONSE:

The binding of purified Pt-IgG and Ctrl-IgG was assessed with transfected HEK cells expressing human IgLON5 and histologically with rat tissue. To confirm the specificity of Pt-IgG, the antibodies were immunoabsorbed with live HEK293 cells expressing IgLON5 or with mock transfected cells. After six rounds of immunoabsorption, the antibodies were applied to sections of rat brain and the reactivity developed as described. Specific immunoabsorption with IgLON5 totally abrogated the reactivity meanwhile was maintained after mock absorption. This method is now presented in detail in the revised manuscript.

COMMENT #R2.12

Please present p-values, asterisk in graphs, and statistical values in tables.

RESPONSE:

We added symbols to indicate statistical significance in Figure 6 and 7 of the revised manuscript and added 11 supplementary tables reporting statistical results, removing the original Table A1.

COMMENT #R2.13

Pages 6,11, Figures: Please use pointers.

RESPONSE:

We included pointers to the details that we wished to highlight in the figure reporting neuropathological results (Figure 8 of the original manuscript; Figure 1 of the revised manuscript). The figure was completely restructured in response to comment 12 of Reviewer 3.

COMMENT #R2.14

Pages 4-7, Discussion: Please discuss the previous studies and present study, weaknesses or limitation in the present study, potentials and significance of this study, the ultimate goal, research or knowledge needed to achieve, the future research direction, and the biggest challenge in this goal, among others.

RESPONSE:

We extensively revised the Discussion section to emphasize the points mentioned in this comment. In particular, we added an in-depth discussion of the study limitations, which also highlighted challenges and future research directions. As remarked above, to our knowledge, there are no previous in-vivo studies on animal models of anti-IgLON5 disease that our study can be compared with.

COMMENT #R2.15

Pages 14-16, References: I recommend authors to use more evidence to back their claims, preferably more than 50 references for original articles. Thus, I recommend the authors to attempt to deepen the subject of their manuscript, as the bibliography is too concise: nonetheless, in my opinion, less than 50 articles for a research paper are really insufficient. Currently authors cite only 39 papers, and they are few. Therefore, I suggest the authors to focus their efforts on researching relevant literature: I believe that adding more studies will help to provide better and more accurate background to this study.

RESPONSE:

The conciseness of our bibliography was partly because IgLON5 disorder has been discovered relatively recently, with no published evidence prior to 2014. Nevertheless, we followed the suggestion to elaborate on the topic of our manuscript and to make the bibliography less concise. As a result, the bibliography of our revised manuscript lists 47 papers.

Reviewer 3 Report

In this study, Alvente, Matteoli et al. provide preliminary data on a passive transfer mouse model of potentially pathogenic human IgLON5 IgG. The authors infused patient IgG for 14 days intracerebroventricularly in female WT, and female and male hTau mice, and assessed phenotypic characteristics associated with clinical manifestiation as well as neuropathology. They found trends recapitulating anti-IgGLON5 antibody disease in subgroups of animals.

The study is overall well performed and nicely described. The data show trends in clinical manifestations and do not provide a major increment above previous knowledge. Nevertheless, I consider the study useful for establishing animal models in this condition, provided the manuscript is improved in different ways. Overall, I would rather focus on the neuropathological findings considering the preliminary nature of the study, and acknowledging the difficulty of recapitulating clinical manifestations of a heterogeneous disease, involving sleep-related disturbances, in mice.

I have the following general comments:

  • Although some neuropathological changes are observed using phosphorylated Tau staining, a more comprehensive characterization of neurodegeneration and neuroinflammation would be interesting, as well as the distribution of human IgG in the mouse brains and the effects on IgLON5 (surface) expression. In light of the overall negative behavioural results, these data would provide the basis for further improved models in this disease. The statistical analyses for the neuropathological assessment appear overly complicated. A more conventional quantification method would be more informative.
  • Given the overall negative results and observed trends , the word “significant” should be used more carefully and only if it is supported by the data (see below for more details). While a summary table with the results is appreciated, Table A1 is difficult to interpret, and descriptive statistics (medians/means etc) should be included. In addition, statistical significance (or a lack thereof) should be clearly indicated in the Figures.

Further specific comments:

Methods

  • Ethical approval for the use of human IgG and conduct of animal studies should also be stated in the methods section.
  • Patient/control characteristics should be described in more detail. How many serum samples were used? What are the disease characteristics of the patients? Are control samples matched for age and gender?

Results

  • Cross-reactivity of Pt IgG to other neuronal/CNS targets should be excluded.
  • Fig 2 could be moved to the supplementary information
  • Fig 6C: a decrease in apneas in Ctrl IgG in REM sleep time is observed. The authors should provide a possible explanation for this finding and whether this may be relevant for the interpretation.
  • The authors state “A significant increase in ventilatory period during sleep was found in female WT and hTau mice treated with Pt-IgG vs Ctrl-IgG at d14 and d45 (P = 0.011, Figure 6D-E)” and “Data in female WT and hTau mice at d14 and d45 indicated a significant decrease of the inter-lick interval associated with Pt-IgG (P = 0.015; Figure 7B-C)”, however in Figs 6D-E and 7B-C no statistical significance is indicated (see also general comment); it appears that statistical significance is reached only after pooling days 14 and 45, therefore the relevance of these results is questionable. In order to conclude statistical significance, it must be demonstrated at a single time point. As also no statistical significance was seen in other subgroups, I would recommend to acknowledge the observed reproducible trends, but not to overstate the results.
  • Fig 8: I recommend extending this figure and consider moving the whole section on neuropathology further up in the manuscript (e.g. as Fig 2 or 3; see general comment). Representative examples with wt animals could be provided as well.

Author Response

We thank the Reviewer for thoughtful comments and criticism. We carefully and extensively revised our manuscript in line with the Reviewer’s requests and suggestions. Please find below our point-by-point responses to the Reviewer’s comments.

COMMENT #R3.1

The study is overall well performed and nicely described. The data show trends in clinical manifestations and do not provide a major increment above previous knowledge. Nevertheless, I consider the study useful for establishing animal models in this condition, provided the manuscript is improved in different ways. Overall, I would rather focus on the neuropathological findings considering the preliminary nature of the study, and acknowledging the difficulty of recapitulating clinical manifestations of a heterogeneous disease, involving sleep-related disturbances, in mice.

RESPONSE:

We thank the Reviewer for the appreciation of our work and of its potential usefulness for establishing animal models of anti-IgLON5 disease. We extensively revised our manuscript, focusing on neuropathological findings, considering the preliminary nature of our pilot study, and acknowledging the difficulty of recapitulating clinical manifestations of a heterogeneous disease such as anti-Iglon5 disease.

I have the following general comments:

COMMENT #R3.2

Although some neuropathological changes are observed using phosphorylated Tau staining, a more comprehensive characterization of neurodegeneration and neuroinflammation would be interesting, as well as the distribution of human IgG in the mouse brains and the effects on IgLON5 (surface) expression. In light of the overall negative behavioural results, these data would provide the basis for further improved models in this disease.

RESPONSE:

We acknowledged the lack of a comprehensive characterization of neurodegeneration and neuroinflammation, of the distribution of human IgG in the mouse brains, and of the effects on IgLON5 (surface) expression among the limitations of our pilot study in the revised Discussion section. We also emphasized that these investigations should be included in future work to develop animal models of this disease. In response to this comment and to comment #5 of Reviewer 1, we checked for the presence of IgG deposits on the brain sections of the mice under study, and the result was negative. This was expected because we euthanized mice for brain harvesting very long after the end of the antibody infusion period to account for possible delayed effects of the antibodies. In particular, brain tissue was obtained approximately 3 weeks after the end of antibody infusion from the 5 mice in which experiments were interrupted due to partial detachment of the head implant, and at least one month after the end of antibody infusion in the other mice. Nevertheless, as just mentioned, this is now highlighted as a limitation of our pilot study, together with the suggestion to explore the kinetics of IgG clearance from the brain after the end of intracerebroventricular infusion in future experiments.

COMMENT #R3.3

The statistical analyses for the neuropathological assessment appear overly complicated. A more conventional quantification method would be more informative.

RESPONSE:

Our pilot study included a preliminary neuropathological assessment whereby three blind researchers attempted to classify the mice based on the extent of p-Tau deposition at immunohistochemistry. In female mice, this classification corresponded quite well to the type of antibodies (anti-IgLON5 patient vs control subject) that were administered to the mice, and we applied standard statistics (Chi square and Cohen’s kappa) to show that this correspondence was unlikely to have resulted from chance. In line with this comment, we removed the statistical analysis of neuropathological data from the revised manuscript.

COMMENT #R3.4

Given the overall negative results and observed trends, the word “significant” should be used more carefully and only if it is supported by the data (see below for more details).  

RESPONSE:

In line with this comment and with comment #11 by this Reviewer below, we focused the revised manuscript on a qualitative description of the results, acknowledging the observed reproducible trends.

COMMENT #R3.5

While a summary table with the results is appreciated, Table A1 is difficult to interpret, and descriptive statistics (medians/means etc) should be included. In addition, statistical significance (or a lack thereof) should be clearly indicated in the Figures.

RESPONSE

In line with this comment and with comment #12 of Reviewer 2, we removed the original table A1, added 11 supplementary tables reporting statistical results and descriptive statistics, and clearly indicated statistical significance in Figures 6 and 7 of the revised manuscript by adding symbols and modifying figure captions. We found that inclusion of the descriptive statistics to the new statistical tables made them cumbersome and significantly detracted from their readability, which risked replicating the complexity that characterized Table A1 in the original manuscript. Moreover, descriptive statistics in tables would essentially duplicate information in the graphs, which show single data points overlaid to summary values. For these reasons, we preferred to include in the revised manuscript supplementary tables with statistical results only. If the Reviewer is not satisfied with this choice and with our reasons for it, we will further modify the manuscript to include also descriptive statistics in the supplementary tables.    

Further specific comments:

Methods

COMMENT #R3.6

Ethical approval for the use of human IgG and conduct of animal studies should also be stated in the methods section.

RESPONSE:

The revised Methods section has been updated as suggested.

COMMENT #R3.7

Patient/control characteristics should be described in more detail. How many serum samples were used? What are the disease characteristics of the patients? Are control samples matched for age and gender?

RESPONSE:

As also mentioned in our response to comment #9 of Reviewer 1, the patient IgGs that were infused to the mice in our pilot study were purified from a single patient. The patient, a female aged 65 years, was very typical clinically, with predominant bulbar phenotype and chronic evolution, positivity to HLA susceptibility alleles, and IgG4 predominance. The control IgG were purified from a control subject matched for age and gender. This is now clarified in revised Methods section, and the possibility that patient features influenced the results is discussed in the revised Discussion section.

Results

COMMENT #R3.8

Cross-reactivity of Pt IgG to other neuronal/CNS targets should be excluded.

RESPONSE:

As also mentioned in our response to comment #4 of Reviewer 1, the IgGs were tested in vitro not to include extra reactivities by immunoabsorption before injecting the animals. This is now clarified in the revised Results section. The revised Methods section was modified accordingly. In particular, the binding of purified Pt-IgG and Ctrl-IgG was assessed with transfected HEK cells expressing human IgLON5 and histologically with rat tissue. To confirm the specificity of Pt-IgG, the antibodies were immunoabsorbed with live HEK293 cells expressing IgLON5 or with mock transfected cells. After six rounds of immunoabsorption, the antibodies were applied to sections of rat brain and the reactivity developed as described. Specific immunoabsorption with IgLON5 totally abrogated the reactivity meanwhile was maintained after mock absorption.

COMMENT #R3.9

Fig 2 could be moved to the supplementary information

RESPONSE:

Done.

COMMENT #R3.10

Fig 6C: a decrease in apneas in Ctrl IgG in REM sleep time is observed. The authors should provide a possible explanation for this finding and whether this may be relevant for the interpretation.

RESPONSE:

This difference in sample means was not supported statistically. This is now clarified in the revised Results section.

COMMENT #R3.11

The authors state “A significant increase in ventilatory period during sleep was found in female WT and hTau mice treated with Pt-IgG vs Ctrl-IgG at d14 and d45 (P = 0.011, Figure 6D-E)” and “Data in female WT and hTau mice at d14 and d45 indicated a significant decrease of the inter-lick interval associated with Pt-IgG (P = 0.015; Figure 7B-C)”, however in Figs 6D-E and 7B-C no statistical significance is indicated (see also general comment); it appears that statistical significance is reached only after pooling days 14 and 45, therefore the relevance of these results is questionable. In order to conclude statistical significance, it must be demonstrated at a single time point. As also no statistical significance was seen in other subgroups, I would recommend to acknowledge the observed reproducible trends, but not to overstate the results.

RESPONSE:

As mentioned also in the response to comment #12 of Reviewer 1, these statements referred to the ANOVA main effect of the anti-IgLON5 antibodies, which is evaluated statistically by taking account of data at d14 and d45. To the best of our knowledge, analysis of ANOVA main effects is a standard and valid statistical approach. We did not test simple effects at single time points in the absence of significant antibody (IgLON5 vs control) x time (d14 vs d45) ANOVA interactions, so as to limit the number of statistical tests we performed, as explained in our response to comment #7 of Reviewer 1. Nevertheless, we understand that our approach may have been confusing and risked overstating the results. Thus, as suggested, we rephrased the Abstract, Results, and Discussion section to acknowledge the observed reproducible trends without overstating the results.

COMMENT #R3.12

Fig 8: I recommend extending this figure and consider moving the whole section on neuropathology further up in the manuscript (e.g. as Fig 2 or 3; see general comment). Representative examples with wt animals could be provided as well.

RESPONSE:

As suggested, we extended the figure on neuropathology, including representative examples with WT mice, and moved the whole section on neuropathology up in the revised manuscript.

Round 2

Reviewer 1 Report

I thank the authors for their revisions. I think the changes are satisfactory.

Author Response

We thank the Reviewer for his/her appreciation of our work.

Reviewer 2 Report

25 February 2022

Review on the manuscript titled “Pilot study of the effects of chronic intracerebroventricular in fusion of human anti-IgLON5 disease antibodies in mice” by Alvente S et al., submitted to Cells

Manuscript ID: cells-1590917

Dear Authors,

Anti-IgLON5 disease is a rare neurodegenerative autoimmune disease, characterized by parasomnias and chorea, associated with antibodies against a cell-adhesion protein IgLON5, and related to the accumulation of phosphorylated Tau protein in the brain stem. The authors studied the clinical and pathological consequence of passive intracerebroventricular immunization of antibodies from patients with Anti-IgLON5 disease in mice. The results showed decreases in breathing rate during sleep and interlick interval during wakefulness and expression of phosphorylated Tau proteins in female wild type and female human Tau protein expressing transgenic mice, but not in male mice of both types. The authors concluded larger studies are necessary to reveal sex differences in mice.

The authors addressed their response adequately and the manuscript is revised accordingly. The manuscript contains seven figures, no table and 47 references. The manuscript carries important value trying to present in vivo evidence of contribution of sex factor in the pathogenesis of Anti-IgLON5 disease. I recommend this manuscript for publication in current form.

Author Response

We thank the Reviewer for his appreciation of our work.

Reviewer 3 Report

The revised manuscript has improved considerably and now does acknowledge its limitations more clearly. The authors now put more emphasis the neuropathological findings considering the preliminary nature of the study, also acknowledging the difficulty and complexity of evaluating the results of administration of human antibodies to experimental animals. Conclusions are supported by qualitative description of results.

Minor comments:

77 and 433: We aimed to perform for the first time a pilot study …

A pilot study implies that it is done the first time. For readability I suggest removing “for the first time”

288: maintained instead after mock absorption with non-transfected cells.

These results are not shown, which should be indicated

304: please indicate whether C, D are wt or hTau mice

305: star should be asterisk

Author Response

We thank the Reviewer for his/her appreciation of our work and for constructive comments. We further revised our manuscript in line with the Reviewer’s suggestions. Please find below our point-by-point responses.

COMMENT

77 and 433: We aimed to perform for the first time a pilot study … A pilot study implies that it is done the first time. For readability I suggest removing “for the first time”

RESPONSE

Done.

COMMENT

288: maintained instead after mock absorption with non-transfected cells. These results are not shown, which should be indicated

RESPONSE

Done.

COMMENT

304: please indicate whether C, D are wt or hTau mice

RESPONSE

Panels C and D of Figure 2 show brain sections from female hTau mice. This is now indicated in the figure caption.  

COMMENT

305: star should be asterisk

RESPONSE

The figure caption has been modified as suggested.